# Cost Evaluation of Professional Services in a Rural Community Pharmacy: A Monocentric Exploratory Approach

**DOI:** 10.3390/pharmacy11050156

**Published:** 2023-09-26

**Authors:** Luis A. Martínez, Cristina García, Lucrecia Moreno

**Affiliations:** 1Community Pharmacist, 02161 Albacete, Spain; 2Department of Medical Sciences, School of Pharmacy, University of Castilla La Mancha (UCLM), 02171 Albacete, Spain; 3Cátedra DeCo MICOF-CEU UCH, University Cardenal Herrera-CEU, 46115 Valencia, Spain; 4Department of Pharmacy, University Cardenal Herrera-CEU, 46115 Valencia, Spain

**Keywords:** community pharmacy, professional pharmacy services, cost analysis, multicompartment compliance aid, medicine dispensing

## Abstract

The increasing pressure on healthcare systems (HCSs) is a cause for concern worldwide. Rising costs, uncertainty about sustainability, and aging populations are the main issues that make it challenging to allocate scarce resources to the needs of HCSs. Clinical professional pharmacy services (PSs) have been shown to help alleviate system stress and to reach the entire population, although a cost of provision is borne. The objective of this study was to evaluate the provision costs of three PSs, a medicine-dispensing service (MDS), a multicompartmental compliance aid system service (MCAS), and a cognitive impairment screening service (CISS), in a rural community pharmacy. A cost analysis was performed using a time-driven activity-based costing model. The time dedicated to PS provision was appropriately recorded, and the corresponding expenses were extracted from the accounting records. A provision time of 4.80 min and a cost of EUR 2.24 were estimated for the MDS, while 18.33 min and EUR 8.73 were calculated for the MCAS, and 122.20 min and EUR 56.72 were calculated for the CISS. The total provision time represented 85% of the pharmacist’s effective working time. Tailored cost analysis is a useful tool for making decisions on the implementation of a PS. Larger studies including a variety of pharmacies and locations are necessary to accurately assess costs and engage in discussions on funding and remuneration.

## 1. Introduction

Healthcare systems (HCSs) are facing economic constraints and structural weaknesses. The primary care workload is also steadily increasing, giving rise to a high prevalence of workforce burnout and social unrest in many countries [1,2,3]. In this context, population aging, chronic non-communicable disease management, and high costs associated with low therapeutic adherence emerge as additional stressors. Although other factors are known to be involved, aging stands out as the main driver of the significant increase in the prevalence of chronic conditions and polypharmacy [4], whose handling requires a significant budgetary effort to cope with the rise in healthcare costs. Closely related to this, non-adherence issues, negative outcomes associated with medication (NOM), and drug-related problems (DRPs) may increase the economic pressure [5]. This situation undoubtedly has an impact on HCS management. In Spain, this is especially relevant in already-aged rural areas, where depopulation and barriers to accessing the HCS lead to a more complex scenario.

Starting with the original concept of pharmaceutical care, the role of community pharmacies (CPs) is significantly changing worldwide, moving beyond the mere delivery of medicines to the provision of clinical professional pharmacy services (PSs) [6]. Spain is no stranger to this global trend, wherein the Ministry of Health Consensus on Pharmaceutical Care has started expanding toward the development of a wide range of PSs [7]. At present, several PSs have been defined, protocolized, and drafted by a working group of experts in practical guidelines to facilitate their homogeneous implementation and provision [8]. Evidence of positive health outcomes for patients and cost savings to HCSs for a variety of PSs is available [9,10,11], suggesting their important role in the maximization of therapeutic benefits for patients and the optimization of the cost-effectiveness ratio of the use of medicines. Moreover, the high accessibility of CPs as the first point of contact with the HCS can be crucial in reaching rural dwellers [12], suggesting that the effective and regulated integration of CPs into the front line of healthcare access might be an effective tool to relieve stress on the HCS.

Apart from dispensing, no PS is currently funded by the Spanish National HCS, and their integration into CPs’ daily practice is at the discretion of each pharmacist. The provision of PSs in the community setting has associated economic costs [13], making implementation a compromised decision that must be carefully weighed. An assessment of the economic impact and costs allows providers to properly manage their resources and define a service price, given an eventual funding opportunity. Thus, a prior economic and financial analysis is necessary.

The main objective of this study was to evaluate the economic costs of three PSs in a particular rural CP setting, including (i) a medicine-dispensing service (MDS), (ii) a multicompartmental compliance aid system service (MCAS), and (iii) a cognitive impairment screening service (CISS), to provide information for management decision making on service implementation, resource optimization, and sustainability over time.

## 2. Materials and Methods

### 2.1. Study Design

An observational prospective study was conducted from July to December 2022 in a Spanish CP located in an extremely depopulated rural area. Selected PSs were considered to be composed of individual stages according to Spanish standard operating procedures and the World Health Organization—International Pharmaceutical Federation’s good practice guidelines [8,14]. A complete list of the stages is shown in Table 1. The working time required to complete each stage was recorded following the time-and-motion technique [15]. The different stages of the services were timed by an external observer, and then, averaged. In the case of MCAS, the software recorded the time taken to fill the device. The total duration of the service was also recorded automatically using software in the MDS. The duration of each PS was calculated as the sum of the duration of each of its stages. The rest of the necessary information on the services was retrieved from the invoices and the corresponding management software. Data were collected at random one day every two weeks throughout the study and stored in an ad hoc Excel spreadsheet. The exclusion criteria, if any, are detailed in the section corresponding to each service.

### 2.2. Clinical Professional Pharmacy Services (PSs)

#### 2.2.1. Medicine-Dispensing Service (MDS)

Active MDS provision begins with identifying the patients to whom the medicines are prescribed, ruling out the concurrence of any non-dispensing criteria or administrative issues. In general, a prescription entitles the patient to receive only one pack of a given medicine at a time, although it is possible to dispense prescriptions for different medicines simultaneously. After obtaining a prescription (usually retrieved via electronic means), the pharmacist gathers information and relevant clinical data, checking whether the medicine has been prescribed for the first time and ensuring that the patient knows the necessary information to ensure correct and safe use. The service ends after delivering the medication and performing the necessary administrative tasks to obtain the refund from the national HCS (see Table 1). An MDS may be interrupted upon the occurrence of any critical incidence. Some examples are the detection of a DRP or an NOM and the occurrence of any administrative issue. Cases in which an incidence led to the termination of the MDS were excluded for calculation purposes. Any MDSs that included any parallel activity apart from dispensing (i.e., a minor ailment service) were not considered in the calculations. The MDSs excluded from the calculations represented approximately 15% of the total.

#### 2.2.2. Multicompartmental Compliance Aid Systems Service (MCAS)

The provision of an MCAS comprises recurrent and non-recurrent activities [16]. After enrolment and obtaining informed consent, an initial medication use review and individualized case study of each patient is carried out, including education on usage, and the analysis of medicine splitting and reconditioning possibilities. This initial interview constitutes the main non-recurrent stage of the service. Some steps in the working procedure (changing the patient’s pharmacotherapy or modifying finished devices) may not be delivered weekly and, thus, should also be considered non-recurrent stages.

The information recorded during the interview allows for an individual assessment and the appropriate selection of the device’s content. The medication is stored in individual patient-specific containers after dispensing, and the data matrix codes of each individual pack are recorded and assigned to the corresponding patient using a specific computer program for batch number tracking. Custom device preparation is carried out weekly by withdrawing medications from the corresponding containers. Any unused doses are returned to the patients’ containers and stored securely until necessary. Blisters are labeled and conveniently stored until use. No automation is available in the CP. The disposal of empty packaging and unused medicines is in accordance with Spanish regulations.

#### 2.2.3. Cognitive Impairment Screening Service (CISS)

The provision of a CISS consists of a personal structured interview using a questionnaire comprising a selection of validated screening tests to assess cognitive impairment through several risk and protective factors classified into six categories: (i) sociodemographic factors; (ii) cognitive impairment; (iii) meaning in life; (iv) psychosocial factors; (v) health problems; and (vi) lifestyle. After checking patients’ eligibility, face-to-face appointments were scheduled. Once signed informed consent was obtained, the patients were interviewed at appropriate facilities. Additional tasks for data processing and analysis, which considerably increase the time required per patient, were also considered for the calculations. The details of the whole procedure have been published elsewhere [17].

### 2.3. Cost Analysis

The cost of providing each PS was assessed using the time-driven activity-based costing (TDABC) method [18]. The cost evaluation required the previous calculation of (i) the total PS provision time, which was estimated by adding the time required for each service stage, and (ii) the capacity cost rates (CCRs), a measure of the cost of the stages per unit of time. The cost of each stage was straightforwardly calculated by multiplying the time by the CCR.

The CCRs were calculated based on the cost of each stage and their effective pro-vision time. The total costs were estimated considering operating expenses divided into (i) provider costs, (ii) support/maintenance costs, and (iii) PS-specific costs. Provider costs comprise a pharmacist’s gross salary plus mandatory taxes and were obtained from the 2022 Collective Agreement for Spanish Community Pharmacies and the National Health Insurance report, respectively [19,20]. Support/maintenance costs are common expenditures attributable, on a pro rata basis, to all the stages and services and were obtained from the pharmacies’ accounts. PS-specific costs, those attributable exclusively to a given service, were individually allocated to each stage and also obtained from the original invoices. Capital cost, initial investment, storage space, and depreciation-derived costs, which do not appear in our accounting records, were not considered in our calculations (see Table 2 for the cost category list).

A pharmacist’s total working time per year was considered regulated in the existing agreements, considering an assistant pharmacist as the service provider (1783 h per year). A non-effective working paid time of 91 h per year was considered to assess the practical capacity [19]. The CCRs, expressed in EUR/min, were calculated by dividing the total cost of each stage by the effective working time spent in the provision.

## 3. Results

The PS provided to the greatest number of patients was the MDS, while the average age of the patients was the highest for the MCAS. In all cases, most of the patients were women, ranging from 55% to 65%. Some characteristics of the PSs and of the target population are shown in Table 3.

The average time for MDS provision was 4.80 min. Prescription processing, together with medicine selection, accounted for ca. 50% of the duration, while an average of 1.8 prescriptions were dispensed per pharmacist–patient interaction. The CISS was the most time-consuming PS per individual unit (122.20 min), with the questionnaire administration stage accounting for ca. 60% of the provision. Since the MCAS consists of recurring and non-recurring stages, estimating the overall service duration by adding the time spent at each individual stage is inadequate, as service variations lead to different resource demands. Thus, the average time of the MCAS was calculated using a time equation representing the time required to deliver recurrent stages plus the incremental time associated with each possible variation [21]. In this way, an average provision time of 18.33 min per device was calculated, with filling being the most time-consuming of the weekly stages (6.80 min, approximately 40%). The initial interview, a non-recurrent stage, was the longest-lasting phase, consuming ca. 40 min (approximately 70%) of the full delivery time of the enrolment stage. The time spent per individual stage of the PS is shown in Appendix A.

The total time spent on the assessed PSs amounted to 1437 h per year. This represents ca. 85% of the theoretical–practical capacity (effective working time), indicating that the remaining 15% is still available for other pharmacy activities. Table 4 summarizes the average PS time, the extrapolated PS annual time per patient, and the corresponding percentage of the practical capacity spent by each PS.

Regarding costs, provider expenses varied from EUR 37,853.91 per year, in the case of an associate pharmacist, to EUR 24,787.81, in the case of a technician. The support/maintenance costs were EUR 9054.48 per year, while the specific costs per year amounted to EUR 1333.78 for the MDS, EUR 2197.48 for the MCAS, and EUR 578.50 for the CISS. The costs of the PSs calculated for an associate pharmacist using the CCRs of the corresponding stages are summarized in Table 5.

## 4. Discussion

As a rule, the largest contribution to the cost of any of the assessed PSs was associated with the salary (plus taxes) of the provider, accounting for ca. 80% of the overall costs. Consequently, the variation in the calculated CCR for each service stage was small. Therefore, if these services were delivered by a technician, the CCR would decrease by 30%, which represents a significant global cost saving. Although this approach may be interesting from an economic point of view, it could lead to organizational issues since, according to Spanish regulations, supervision by a pharmacist is necessary in some stages of provision (entailing an additional cost). Conversely, in the case of a pharmacy owner, the cost of the provider can be difficult to estimate, and it is subject to significant variability depending on the annual net profit of the CP. Thus, all costs were calculated assuming that the PSs were completely provided by an associate pharmacist.

The average time required to provide the MDS (4.80 min.) is consistent with the existing literature data. Previous estimations in Spanish and Portuguese CPs are slightly lower (4.16 and 4.41 min, respectively) [22,23]. The time required for post-dispensing prescription handling and invoice processing, which were considered in the calculations, may explain the observed differences. Regarding costs, the results (EUR 2.24) are practically identical to those available in the literature [23,24].

In Spain, the CP revenues from the National HCS are linked to the tax-free price of the dispensed medicine. Since the fixed fee for dispensing is equivalent to 27.9% of this price, it is possible to estimate that the MDS will cover its costs only if the price of the dispensed medicine is above EUR 8.39. Statistical data provided by the Spanish Ministry of Health indicate that the average price of a dispensed medicine in Albacete during 2022 was EUR 11.44, which confirms that the MDS is a profitable service [25]. Additional sources of income for Spanish CPs are a fixed commercial margin on non-prescription medicines and the sale of OTC products. Although there is considerable variation between CPs, this represents less than 8% of the total income of a rural CP (6% in our specific scenario). Therefore, the benefits obtained from dispensing are the main source of income for a rural CP, making sustainability still possible, although challenging. This is noteworthy since pharmacists practicing in rural areas are health professionals who are responsible for a segment of the disfavored and aged population, which is characterized by high prevalence rates of polymedication and pluripathology and whose pharmaceutical care is the cornerstone of a universal and equally accessible Spanish HCS. In this context, implementing additional services would contribute to rural CPs’ sustainability if they were profitable.

Although non-recurrent in nature, the CISS was the most expensive service per patient, costing approximately EUR 56. Spanish CPs actively collaborate on campaigns aimed at promoting community health. Focusing on those promoted by health authorities, initiatives covering a wide range of services, such as face mask provision during the COVID-19 pandemic, needle-syringe exchanges, and methadone supplies for drug addicts, can be mentioned [26]. However, the participation of CPs in these initiatives is fully altruistic since they are not funded. In this regard, smoking cessation can be considered the only exception; although no fee per service is paid, pharmacological treatment is funded by the National HCS under certain conditions (one attempt to cease per patient and year and oral therapy [27]). Therefore, the provision of an expensive community health service with no budgetary allocation can only be carried out if an alternative source of financing is available. In our case, the CISS was provided as a part of Ph.D. research (see the acknowledgments), and once its purpose was fulfilled, the service was not sustainable over time. With this in mind, developing campaigns of this magnitude (i.e., involving the screening of a health problem) is a difficult undertaking for Spanish CPs as it involves many actors and requires careful designing and planning.

The MCAS is a complex, individualized process that must be analyzed in depth for each patient. Since it is a post-dispensing service, the MCAS does not depend on the price of the medicines. The literature on positive health outcomes linked to MCAS use in the community is scarce. Even so, there is enough evidence to consider the MCAS as a tool that may contribute to solving the complex problem of adherence [28,29,30]. In Spain, recent studies have proposed MCAS use as a cost-effective, user-friendly tool for blood pressure and cholesterol level control [31,32]. Regarding costs, Rius et al. calculated a cost of EUR 19.85 per patient per month in a Spanish CP in 2012 [33], 90% of which corresponded to the cost of the provider (16.75 EUR/h). We estimated a cost of EUR 37.73 per patient per month without a commercial margin. For comparison’s sake, if we considered only the expenditures due to the salary of the pharmacist (22.37 EUR/h in 2022), the MCAS’s estimated cost would decrease to approximately EUR 29, which is similar to Rius’s assessment in 2012, correcting costs by the difference in the consumer price indexes.

Some Spanish local administrations have shown interest in the MCAS, and a few initiatives in which the cost of the service is reimbursed to the pharmacist are already implemented. In Albacete, a collaboration agreement between the Provincial Government and the local Association of Pharmacists, in force since 2021, makes it possible for ca. 240 patients located in rural depopulated areas to receive MCASs through 22 different CPs. Under this agreement, each pharmacist is currently receiving EUR 22.37 per patient and month, plus all specific costs and a mileage fee. According to our estimations, excluding specific expenses would decrease the cost to ca. EUR 36 per patient per month, which is above the remuneration embodied in the agreement. Assuming that no cost should be passed on to patients, an additional budgetary effort appears necessary to (i) allow full-service costs to be covered with a certain profit margin for the CP and (ii) ensure service provision to all eligible patients. Otherwise, the consolidation and long-term sustainability of the service and health outcomes cannot be achieved [34].

Data on the evaluation of the costs and the provision times of other PSs by Spanish community pharmacists are also available in the literature. It has been calculated that pharmacists spent 11 h per year providing medication reviews with follow-ups per patient, resulting in a calculated price of EUR 22 per patient-month, including a 30% profit margin. The average time required to provide a minor ailment service has been estimated at ca. 5 min, while the service cost is EUR 2.13, excluding medication price [35,36,37]. Considering the overall data according to PS classification [8], services to improve the medicine utilization process have a lower cost but are provided to a larger number of patients, accounting for a significant part of the working time. Meanwhile, PSs aimed at achieving health outcomes and improving adherence are more costly, despite reaching fewer patients, thus requiring a lower percentage of practical capacity. In this study, the MCAS was found to be as time-consuming as the MDS, pointing to convenient funding as a key aspect of its sustainability over time.

Some methodological limitations must be mentioned. Firstly, this study was conducted in a single community pharmacy, limiting the generalizability of the calculations; the heterogeneity of Spanish CPs’ characteristics requires a multicentric approach if the global situation is to be evaluated. However, a monocentric study is not without utility, as it provides a custom analytical tool to easily improve the management of a CP by optimizing costs and times in daily routines. Secondly, the measurement of the duration of the services was a very time-consuming task. Moreover, the observer in charge of time recording was familiar with the working procedures, which may be a source of bias. Thirdly, the data were recorded while a single worker, an assistant pharmacist, provided all the monitored PSs during this study. This may also constitute a source of bias since PSs are jointly provided by all workers during real daily work routines.

## 5. Conclusions

Only remunerated PSs are sustainable in the long term. Funding is a crucial factor especially for CPs that are based in depopulated areas. With this background, the MCAS represents a good alternative for implementation in a rural CP since it brings together cost-effectiveness studies supporting positive health outcomes for patients, cost savings to the HCS, and gained experience in funding. Larger studies that include data from a variety of CPs are necessary to accurately assess costs, as this is essential before deciding whether to implement a PS. Moreover, a solid economic study may be useful to engage in discussions on global funding and remuneration with public or private entities. Once the costs have been defined, it is important to consider the application of a commercial profit margin. Otherwise, the provision of the MCAS will not be profitable and will therefore not contribute to the sustainability and long-term economic viability of the CPs.

## Figures and Tables

**Table 1 pharmacy-11-00156-t001:** Stages of the clinical professional pharmacy services selected in this study.

**Medicine-Dispensing Service**
1. Patient identification and obtaining prescription(s)
2. Collection and evaluation of clinical information
3. Medicine selection
4. Prescription processing
5. Medicine delivery with guidelines for safe and correct use
6. Payment
7. Post-dispensing prescription handling and invoice processing
**Multicompartmental Compliance Aid System Service**
**Recurrent stages**
1. Device preparation
a. Medicine dispensing
b. Patient’s personal medicine container filling
c. Device filling
d. Device content checking, sealing, and labeling
2. Invoice processing
**Non-recurrent stages**
3. Service enrolment
a. Service offering, informed consent, and initial appointment
b. Interview
i. Personal, clinical, and pharmacological data gathering
ii. Medicine use review
iii. Data collection form filling
c. Agreement on the device’s content
d. Data transcription to management software
e. Informed report to primary care physician and nurse
4. Change in pharmacotherapy
a. Data modification
i. Medicine use review
ii. Data collection form filling
b. Agreement on the device content
c. Transcription to management software
5. Modification of finished devices
a. Opening and discarding the finished device
b. Preparing new device according to modifications
**Cognitive Impairment Screening Service**
1. Service offering, informed consent, and initial appointment
2. Questionnaire administration
3. Data transcription to computer database
4. Data formatting, processing, and storage
5. Informed referral to primary care physician

**Table 2 pharmacy-11-00156-t002:** Costs considered for the capacity cost rate calculation.

**Provider Costs**
1. Pharmacist’s gross salary
2. Mandatory taxes payable by the employer
**Support/Maintenance costs**
1. Utilities: electricity, water, heating, and internet
2. Outsourced services
a. Occupational risk prevention
b. Compliance with personal data protection requirements
c. Security
d. Fire protection
e. Database subscriptions
3. Insurances and taxes
**Specific costs**
1. Software licensing and updates
2. Equipment materials (blisters, protective clothes, questionnaire booklets, etc.)
3. Continuous training

**Table 3 pharmacy-11-00156-t003:** Characteristics of the studied services and the attended population.

Medicine-Dispensing Service	N (%) or Mean (DE)
Patients	380 (100)
Gender	
Female	210 (55)
Male	170 (45)
Age	55.5 (23.0)
0–20	39 (10)
21–50	87 (23)
51–80	197 (52)
>80	57 (15)
Number of prescriptions	6611
Number of dispensations	3698
**Multicompartmental Compliance Aid System Service**	
Patients	39 (100)
Gender	
Female	25 (64)
Male	14 (36)
Age	80.3 (8.3)
<60	2 (5)
61–75	6 (15)
76–90	30 (77)
>90	1 (3)
Number of devices	984
Reconditioned medicines per device	6.9 (2.6)
Service enrolment	5
Changes in pharmacotherapy	25
Modification of devices	24
**Cognitive Impairment Screening Service**	
Patients	60 (100)
Gender	
Female	38 (63)
Male	22 (37)
Age	
51–65	12 (20)
66–80	26 (43)
>80	22 (37)
Cognitive impairment prevalence	24.4%

**Table 4 pharmacy-11-00156-t004:** Average time per service, extrapolated annual time per patient, and percentage of the practical capacity of each studied service.

Service	Average Time (min) ^¥^	Extrapolated Time (Hours/Patient Year)	Practical Capacity (%)
Medicine dispensing	4.80	1.56	35.0
Multicompartment compliance aid system	18.33	15.42	35.5
Cognitive impairment screening	122.20	2.04	14.4

^¥^: for details on time spent in service stages, see Appendix A.

**Table 5 pharmacy-11-00156-t005:** Average service costs per unit and per patient a month.

Service	Cost per Unit (EUR)	Cost per Patient (EUR/month)
Medicine dispensing	2.24	3.63
Multicompartment compliance aid system	8.73	37.83
Cognitive impairment screening	56.72	9.45

SD: standard deviation. Max: maximum. Min: minimum.

## Data Availability

The data used for this study are available upon request.

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
