# Peer review of "Cost Evaluation of Professional Services in a Rural Community Pharmacy: A Monocentric Exploratory Approach"

_pharmacy, 2023, doi:10.3390/pharmacy11050156_

Round 1
Reviewer 1 Report
Thank you for the opportunity to review this paper. I found it very interesting and have only some minor comments for the authors – listed below:
- 2.2.1. Medicine-dispensing service (MDS) - I do not understand why the cases with DRP or NOM detection were excluded for calculation purposes. As they may happen (and sometimes do) they should be included in calculations because detecting them is actually the purpose of the service so the time (and costs) pharmacists spend on them should also be taken into account. Please elaborate on your rationale.
- Limitations – “Secondly, the data collection was a very time-consuming task.” - was this supposed to be another limitation (which I do not understand) or explanation of the first limitation? As you generally seem to start the limitations by firstly, secondly and thirdly I suggest rephrasing to remove doubts
- Please put conclusions under a separate section and elaborate some more on practical implications of your study
Author Response
File attached

Reviewer 2 Report
The authors have tackled a very important issue in an original and enlightening way. This well-written manuscript deserves to be published, provided that more details are given to improve repeatability in different contexts.
Here are the main points for improvement:
- Provide more raw data in an appendix or online repository (e.g. subset data about dispensing).
- Explain in more detail how the data was collected, so that other research groups can replicate this study (especially for the MDS).
- Provide an estimate of the distribution and uncertainty of the results.
- Pay more attention to the sustainability of any intervention. The authors state on line 135 that “No initial investment, storage space, or depreciation-derived costs were considered in the calculations”. This needs to be thoroughly justified. It would be better to provide a rough estimate of these costs rather than ignoring them completely, especially since the local aspect of the PS is so important. At the same time, the holistic economic aspect of the local pharmacy is ignored. What are the other sources of income that make altruistic services possible? This cannot be ignored.
Some minor points:
- Reference 3 is incomplete.
- On line 91, it is stated that interruption of the MDS and parallel activity were reasons for exclusion. We need data on this to assess its importance and influence on the calculation of total activities.
- How much automation was available in the pharmacy (e.g. a robot)?
- Were language and other communication barriers a frequent occurrence in the pharmacy?
- What was the cost or time investment for gathering data on MDS?
Dispensing involves much more than simply moving a box. Any data that can describe this process in more detail would be interesting for policymakers who lack real-world experience and for pharmacists to evaluate their own service.
Author Response
File attached

Reviewer 3 Report
This study reports findings from a single rurally located pharmacy in Spain regarding the costs of providing three clinical professional pharmacy services. Stop-watch evaluations and averaging involved a single pharmacist carrying out these operations. The provision of the services do no appear to be based upon professional guidelines developed for their provision. Capital, and overhead costs and profit margins have not been included in the costing. Activities such as ordering and stock management and a reasonable patient counselling time for the prescription service are not included. Some of the presumably average times appear short. It is also confusing what a prescription was. Was this a single dispensed item or multiple items? It is quicker to dispense a prescription with two or more items per prescription than one with a single item; on a per item time measurement. No time has been allocated for checking with the prescriber. With the MCAS, no time seems to have been allocated to counsel the patient about use. Not all medicines are suitable for inclusion in devices, which can lead to patient confusion. When medicines are changed no patient counselling seems to have been recorded. It is unclear how residual medicines are stored as nothing is mentioned about batch number management for tracing medicines if a recall occurs. Disposal times are rapid. Does this meet environmental requirements. Usually a one week supply does not reflect the amount in a standard pack, which is dispensed and then packed in MCAPs on a weekly basis. No overhead costs including the device are included. The CISS seems to have been included for research purposes rather than a professional service within the normal scope of a community pharmacist. Often specialised qualifications are required to administer such a service. No justification is provided for this service and what referral pathways were available.
This study appears to have overlooked a range of capital costs, overhead costs, profit and down-time. It seems to desire to report minimised costs which if used as a template could make service provision using these costings rendering a business unviable. The study needs to be widened to a representative sample of pharmacies and the full underlying costs of providing these services included.
Some words need to be reconsidered such as "obtention" Page 3 Line 99.
Author Response
File attached

Round 2
Reviewer 3 Report
The authors have responded to the questions raised initially and have addressed issues where possible. The study still suffers from it being a single rural community pharmacy, limiting the generalisability of the data reported. A minamilist approach using associate pharmacist salaries or even a technician, may also not reflect costs in other similar pharmacies. Payers often which to also minimise their costs and this paper may provide evidence to help with that.